# Numerical Analysis of the Heating Characteristics of Magnetic Oscillation Arc and the Fluid Flow in Molten Pool in Narrow Gap Gas Tungsten Arc Welding

**DOI:** 10.3390/ma13245799

**Published:** 2020-12-18

**Authors:** Xiaoxia Jian, Xing Yang, Jingqian Li, Weihua Wang, Hebao Wu

**Affiliations:** 1School of Mechanical and Electrical Engineering, Wuhan Institute of Technology, Guanggu 1st Road, Wuhan 430205, China; yx2020wit@163.com (X.Y.); lijinggan0@gmail.com (J.L.); 2School of Mechanical and Electrical Engineering Rongcheng Campus, Harbin University of Science and Technology, No. 52 Xuefu Road, Harbin 150080, China; wwh_05106@gmail.com

**Keywords:** magnetic oscillation arc, extra magnetic field, narrow gap GTA welding, numerical simulation

## Abstract

Magnetic oscillation arc (MOA) technology was developed to avoid insufficient fusion defects appearing at the sidewalls in narrow gap gas tungsten arc welding (NG-GTAW). In this work, a unified model was developed to simulate the process of MOA assisted NG-GTAW. The model included the MOA, welding pool, workpiece and the coupling interaction between them. The heating characteristic of the MOA and the flow of liquid metal were simulated, and the mechanism of forming a uniform welding bead under MOA was investigated. It was found that if the magnetic flux density increased to 9 mT, the MOA could point to the sidewall directly; the maximum heat flux at the bottom declined by almost half and at the side, it increased by more than ten times. Additionally, the heat flux was no longer concentrated but dispersed along the narrow groove face. Under the effect of MOA, there were mainly two flow vortexes in the molten pool, which could further increase the heat distribution between the bottom, sidewall and corner, and was beneficial for the formation of a good-shape weld. The model was validated by experimental data.

## 1. Introduction

Narrow gap welding is an important joining technology for thick plate structures and is widely used in ship and pipeline manufacturing, the nuclear power industry and so on [1]. Compared with traditional arc welding, in the narrow gap welding process, the gap of the groove is decreased, which can lead to advantages of lower heat input, less filler metal and higher efficiency. However, at the same time, the swing of the welding arc is restricted by the small opening width of the groove, often leading to incomplete fusion defects at the sidewalls of the welding groove [2]. Some narrow gap welding methods have been developed to solve this problem, such as rotating arc [3], swing arc [4] and tandem arc welding [5], and so on. Li et al. applied a multilayer rotating arc for joining medium steel plate [6]. Because the trajectory of the high speed rotating arc approached a circle, they developed a four-point heat source model for the rotating arc and successfully reported the distribution of the temperature field. Dong studied the theoretical foundation of process design for narrow gap gas tungsten arc welding and pointed out that the heat efficiency of the welding arc was different during weaving [7]. The fluid flow of the molten pool in narrow gap welding was also studied experimentally [8] and numerically [9]. Xu et al. used a swing arc to join DH36 steel plates with a thickness of 20 mm. They found that the flow vortex of molten metal under the swing arc was beneficial to increase the sidewall fusion depth [10]. Zhu et al. studied the influence of swing arc parameters on the fluid flow in a molten pool and pointed that the flow pattern of molten metal influenced not only the shape and dimension of the welding beads but also the welding defect formation [11]. These studies showed that by employing the advantages of narrow gap welding methods, the heat flux characteristics of the moving welding arc and the fluid flow of the liquid metal were greatly modified to form well welds.

Compared with mechanical swing and rotating arc, magnetic oscillation arc (MOA) technology is more compact and effective in terms of changing the welding arc behavior and the flow of molten metal. MOA associated narrow gap gas tungsten arc welding (MOA-NG-GTAW) was developed to solve the lack of fusion defects [12] and has been proven experimentally to be effective in obtaining uniform fusion welds [13]. A lot of MOA-NG-GTAW research has been carried out, mainly focused on the improvement of the magnetic field generator [14], the optimization of magnetic field parameters [15], the methods of controlling the weld bead ability [16] and the arc profile with different magnetic field parameters [17,18]. These works made significant efforts to improve the control of this technology; however, the formation mechanism of the uniform welding bead, especially the heat-transfer mechanism, has not been fully investigated.

When an extra magnetic field is added to the welding process, the current flow path, temperature, velocity and other properties of the welding arc and molten pool will be changed [19,20]. Some research results have shown that the temperature field of the welding arc is nonaxisymmetrical under a longitudinal magnetic field [21], and the current density on the workpiece surface is no longer Gaussian distribution if an axial magnetic field is used in the arc column [22]. Until now, studies on the heating and mechanical properties of MOA have been lacking, and the fluid flow of molten metal in MOA-NG-GTAW has not been fully understood, both of which are key factors influencing the formation of the welding bead [23]. An in-depth understanding of these two factors is helpful for better application of this process.

In this paper, we aim to investigate the heating characteristics of the welding arc and the fluid flow of the liquid metal in MOA-NG-GTAW. Because both the MOA and the flow in the molten pool are related to multi-physics coupling, the mathematical simulation method is selected. Under the effect of MOA, the surface of the molten pool surface is changed in real time, and they interact with each other. Therefore, a unified coupling model is developed that includes the MOA, molten pool, tungsten electrode workpiece and their interactions. Finally, the model is validated by experimental data.

## 2. Theoretical Formulation

### 2.1. Computational Domain

Figure 1 shows the principle of MOA-NG-GTAW. The tungsten electrode is fixed with a magnetic field generator, and the welding arc is between the two magnetic poles, as shown in Figure 1a. The electrode is placed with the magnetic generator into the narrow groove of the workpiece, and it is ensured that the magnetic line is parallel to the welding direction. When an alternating exciting current, as shown in Figure 1b, is used, an alternative longitudinal magnetic field (LMF) is added to the welding arc column. The welding arc will be deflected to the sidewall due to the Lorentz force and swings in the narrow gap periodically, as shown in Figure 1c.

MOA-NG-GTA welding can be conducted by the welding system shown in Figure 2a. It includes the welding power, shielding gas, magnetic excitation power, workpiece and welding torch, which is integrated with a magnetic field generator. The argon gas flows into the welding zone from the torch and then flows out from the front and back of the narrow gap; the welding arc burns between the tungsten electrode and the workpiece and swings periodically. In this paper, we mainly pay attention to the MOA and the weld formation, so the welding machine and the magnetic exciting power equipment are not considered. Based on the welding system, the selected simulation domain includes the shielding gas, electrode, welding arc zone, workpiece and LMF, and is shown in Figure 2b. The size of the model was 20 × 20 × 14 mm^3^, the narrow groove was rectangular with a size of 20 × 10 × 9 mm^3^ and the thickness of the workpiece was 5 mm. The welding direction was the positive of the x-axis. The distribution of LMF was assumed uniform throughout the narrow gap. In this model, the process of arc swing and the melting and solidification of the workpiece was simulated, so there were four phases of matter in the simulation domain, which were the shielding gas, arc plasma, molten metal and the solidus workpiece. The shielding gas and the welding arc were treated as gas, the molten metal and the solidus workpiece were treated as fluid phase and the viscosity value of the solidus metal was large enough.

### 2.2. Governing Equations

Because there are electrons, ions, neutral particles and photons in the arc plasma, the thermal and transport properties are difficult to measure. Therefore, several assumptions were made: (1) the arc plasma was in local thermodynamic equilibrium (LTE), and its thermal and transport parameters were only functions of temperature; (2) the fluids in the welding arc and molten pool were incompressible and laminar; (3) the viscous dissipation was not considered. The conservation equations of computational fluid dynamics (CFD) were calculated in the whole domain. The different heat and momentum mechanisms in the welding arc, molten metal and solidus metal were considered by adding different source terms in different zones.

The conservation equations of mass, momentum and energy are as follows:(1)$$\frac{\partial \rho }{\partial t}+\nabla \cdot \left(\rho \vec{V}\right)=0$$
(2)$$\rho \left(\frac{\partial \vec{V}}{\partial t}+\vec{V}\cdot \nabla \vec{V}\right)=-\nabla p+\mu \nabla^{2}\vec{V}+\vec{j}\times \vec{B}+G+{\vec{F}}_{s}$$
(3)$$\rho c_{p}\left(\frac{\partial T}{\partial t}+\vec{V}\cdot \nabla T\right)=\nabla \cdot \left(k\nabla T\right)+\frac{j_{x}^{2}+j_{y}^{2}+j_{z}^{2}}{\sigma }+S_{\varphi }$$
where ρ is the density, *t* is the time, $\vec{V}$ is velocity vector, *p* represents the pressure, *μ* indicates the viscosity, $\vec{j}$ is the current density vector; $\vec{B}$ is the vector of magnetic induction density, *G* is the gravity in z direction, ${\vec{F}}_{s}$ represent the momentum source term, *c_p_* is the specific heat, *k* is the thermal conductivity, *j_x_, j_y_* and *j_z_* represent the current density in *x*, *y* and *z* direction, respectively, σ is the electrical conductivity and $S_{\varphi }$ represents the energy source term.

The Lorentz force produced by the self-magnetic field that is induced by the welding current occurs in the whole domain, and is calculated by the $\vec{j}\times \vec{B}$ term in Equation (2). In the welding arc zone, the momentum source term includes another Lorentz force from the extra LMF, which is expressed as Equation (4). In the molten pool, the additional momentum source term is the buoyancy force, which is expressed as Equation (5).

In the welding arc:(4)$$F_{\mathrm{sg}}={\vec{B}}_{e}\times \vec{j}$$

In the welding pool:(5)$$F_{\mathrm{sl}}=\rho \vec{g}\beta \left(T-T_{ref}\right)$$
where *F_sg_* is the additional momentum source term in the gas phase, $F_{\mathrm{sl}}$ is the additional momentum source term in the liquid phase, ${\vec{B}}_{e}$ is the density of LMF, $\vec{g}$ is the vector of gravity, *β* indicates the thermal expansion coefficient and *T_ref_* is the reference temperature and its value is 300 K in this model.

In the energy conservation Equation (3), the Joule heat exists in the entire domain. In the welding arc zone, the enthalpy transport of electrons and the radiation loss are also considered by Equation (6) [24]. In the weld pool, the latent heat of fusion is calculated by Equation (7).

In the welding arc:(6)$$S_{\varphi g}=\frac{5K_{B}}{2e}\left(j_{x}\frac{\partial T}{\partial x}+j_{y}\frac{\partial T}{\partial y}+j_{z}\frac{\partial T}{\partial z}\right)-U$$

In the welding pool:(7)$$S_{\varphi l}=\frac{\Delta H}{C_{p}}\frac{\partial f_{l}}{\partial t}$$
where $S_{\varphi g}$ is the additional momentum source term in the arc zone, $S_{\varphi l}$ is the additional momentum source term in the welding pool, *K_B_* represents the Boltzmann constant, *e* is the electron charge, *ΔH* represents the latent heat of fusion, *U* is the radiation losses and *f_1_* represents the liquid fraction.

In the above CFD conservation equations, the values of current density and magnetic flux density are needed. The current flows through the workpiece, plasma arc and the electrode, so the Maxwell equations are also solved in the whole domain to calculate the electric field and magnetic field [25].
(8)∇⋅(σ∇ϕ)=0
(9)$$\vec{j}=-\sigma \nabla \phi$$
(10)$$\Delta \vec{A}=-\mu_{0}\vec{j}$$
(11)$$\vec{B}=\nabla \times \vec{A}$$
where *Φ* is the electrical potential, $\vec{A}$ is the magnetic vector potential and $\mu_{0}$ is the permeability of the vacuum.

### 2.3. Boundary Conditions

#### 2.3.1. Interior Boundary Conditions

There were two interior boundaries; one was the interface between the welding arc and the workpiece; the other was the interface between the liquid pool and the solid metal. A sheath layer existed at the former interface where the current was discontinuous, and there was a large gradient in the velocity, temperature and material properties, which was too thin (<2 μm) to be included in the model. The LTE-diffusion approximation method developed by Lowke and Tanaka [26] was applied to deal with the sheath layer. At the same time, the position of this interface changed with the periodical swings of the MOA. Therefore, it was necessary to trace the position of this interface in real time. In this model, the volume-of-fluid method was used to trace the interface between the plasma arc and molten pool [27]. The function of volume fraction (*F*) (Equation (12)) was calculated in the whole domain; the meshes with value 0 < *F* < 1 were at the interface. The mesh size had the smallest value of 0.2 mm near this interface.

The heat fluxes through this interface included the electron condensation heat, the radiation loss and the conduction heat flux, which was calculated by Equation (2); the former two heat fluxes were calculated by Equation (13) and were added to the energy conservation equation as the source term. On the left side of Equation (13), the first term is the electric heat because of the electron absorbed at the interface, and the second term is the radiation loss.

There are four kinds of forces acting on the welding pool surface, which are the arc pressure, drag force from the arc plasma, Marangoni force and the surface tension. The first two terms are calculated by the unified momentum conservation equation, and the surface tension is calculated by the continuum surface force model [28]. The last two terms are calculated by adding the source terms at the interface layer; the Marangoni force *F*_m_ and surface tension *F*_st_ are expressed as Equations (14) and (15).
(12)$$\frac{\partial F}{\partial t}+\vec{V}\cdot F=0$$
(13)$$S_{h}=\left|j_{a}\right|\varphi_{w}-\varepsilon \alpha T^{4}$$
(14)$$F_{m}=\left(\nabla_{S}T\cdot \vec{\tau }\right)\frac{\partial \gamma }{\partial T}$$
(15)$$F_{\mathrm{st}}=\gamma \kappa$$
where *F* is the fraction of fluid volume, *j_a_* is the current density at the anode, *φ_w_* is the work function of workpiece, *ε* is the radiation emissivity, *α* represents the Stefan–Boltzmann constant, $\nabla_{S}$ indicates the free surfaces gradient operator, $\vec{\tau }$ is the unit vector at the tangential, *γ* is the surface tension coefficient and k represents the surface curvature.

There is a mushy zone between the liquid and solid metal in the workpiece. For better modeling of the solidification/melting process, we use the enthalpy-porosity technique to consider the momentum sink and the pull velocity of the solidified material movement happening at the mushy zone. They are calculated by Equation (16), and *F*_mushy_ is added to momentum conservation equation as a source term at the cells at the interface.
(16)$$F_{\mathrm{mushy}}=\frac{{\left(1-f_{l}\right)}^{2}}{\left(f_{l}{}^{3}+\delta \right)}A_{mush}\left(\vec{v}-{\vec{v}}_{p}\right)$$
$$\mathrm{Where},\;f_{l}=\left\{ \begin{array}{l} 0\;\;\;\;\;\;\;\;\;\;\;\;\;\;\;\mathrm{if}\;\;\;T<T_{\mathrm{solidus}}\\ 1\;\;\;\;\;\;\;\;\;\;\;\;\;\;\;\mathrm{if}\;\;\;T>T_{\mathrm{liquidus}}\\ \left(T-T_{\mathrm{solidus}}\right)/\left(T_{\mathrm{liquidus}}-T_{\mathrm{solidus}}\right)\;\;\;\;\mathrm{if}\;\;\;T_{\mathrm{solidus}}<T<T_{\mathrm{liquidus}} \end{array}\right.$$
where *f_1_* represents the liquid fraction, *δ* indicates a small constant to avoid a zero denominator, *A*_mush_ is a constant to measure amplitude damping and ${\vec{v}}_{p}$ is the drag velocity.

#### 2.3.2. External Boundary Conditions

For the external walls of the base metal, the velocity was zero. The electric potential was set as zero; the heat flux at the external boundaries was mainly calculated by the conduction heat and the radiation losses; the groove face was set as interior elements traced by the volume of fluid function. For the tungsten electrode surface, the temperature at the tip was 3000 K, and the current density was set as uniform. The temperature of the sidewall was set as 1000 K, and the electric potential flux here was zero. Table 1 shows the details of the external boundary conditions.

### 2.4. Material Property and Solution Method

Stainless steel SUS 304 was selected as the workpiece material. The specific heat, viscosity and thermal conductivity are functions of temperature [29], and the other properties are shown in Table 2. Pure argon was chosen as shielding gas; its density, thermal conductivity, specific heat, viscosity, electric conductivity and radiation coefficient were functions of temperature [30], and the net radiative emission coefficients came from reference [31]. To make the calculation more stable, at the interface between two phases, the material properties were decided by the software.

Ansys Fluent 11.0 (Ansys, Inc., Canonsburg, PA, USA) was used to calculate the model. The momentum, energy, volume of fluid and solidification/melting modules were used. Because only standard form equations could be calculated, the source terms were added by the user-defined functions, and the Maxwell functions were added by the user-defined scalars. The whole simulation domain was divided into 400,000 hexahedral cells; the minimum mesh size was 0.2 mm, and appeared at the arc column, and the maximum mesh size was 0.4 mm, and appeared at the outside of the workpiece. The SIMPLE solution method was used to calculate the equations (shown in Figure 3). The iteration time step was 4 × 10^−5^ s. When all the residuals of the physical quantity were smaller than 10^−4^, the iteration was regarded as convergent. At the same time, the gas flux report of the domain was also used to verify the convergence. Sliding mesh technology was used to treat the relative movement of the tungsten electrode to the workpiece.

## 3. Results and Discussion

The main welding parameters in the simulation case were as follow: the welding current was 200 A, the electrode diameter was 1.6 mm, the distance between the electrode tip and the groove bottom was 4 mm, the gas flow rate was 25 L/min, the magnetic flux density was 9 mT and the frequency of LMF was 10 Hz.

### 3.1. Behavior of the Magnetic Oscillation Arc

The direction of the Lorentz force acting on the arc plasma produced by the extra LMF is shown in Figure 4. The blue dotted lines are the computational current vector; the black arrow represents the direction of the Lorentz force according to Fleming’s left-hand rule. There were three main directions of the Lorentz force; at the left side, it pointed right down, at the right side, it pointed right up and at the bottom, it pointed to the right. It can be concluded, with the influence of this Lorentz force, the welding arc at one side of the electrode will be compressed, and it will be pushed upward at the other side.

Figure 5 shows the simulated deflecting welding arc at different times. When the density of LMF is 9 mT, the axis of the welding arc is already deflected to the sidewall (except in Figure 5c when the welding arc is in a transient state). The high-temperature arc plasma heats the side directly, and at the same time, the temperature of the arc plasma contacting with the narrow groove bottom is relatively low. Under the MOA, the heating area at the groove bottom is located to one side, and the heating area at the sidewall is almost equal to it at the bottom. Therefore, the temperature distribution characteristics of the MOA are beneficial for the formation of a uniform bead. According to the current wave used in this model, the time for one period is 0.1 s, the first changing time is at 0.025 s and the second changing time is at 0.075 s. Figure 5b–d shows the temperature distribution of the MOA near the changing time. The MOA deflects to the right sidewall till 0.025 s, and at 0.0252 s, i.e., 0.0002 s after the exciting current changing its direction, the welding arc axis has already pointed to the narrow bottom from the sidewall. It can also be seen that the center zone of the welding arc changes faster than the outside region. The deflecting amplitude of the MOA increases to the state before the changing time again at 0.027 s. Thus, the welding arc could potentially finish the transformation of the deflection direction within 0.002 s. During the transition time, the welding arc mainly heats the groove bottom and not the sidewall. However, because the transition time is short, its effect on the formation of welding bead could be omitted, i.e., the frequency of the LMF has little effect on the bead formation, and this point is also verified by experiments [5].

As the welding time increases, the workpiece is melted, and the molten pool is formed, as shown in Figure 5e,f. The surface of the molten pool fluctuates in real time during the deflecting process of the welding arc. As the welding pool surface is depressed, the welding arc profile adjacent to the welding pool is changed correspondingly, and this interaction changes over time. Thus, it is necessary to include the welding arc in the model.

As stated in Section 2.3.1, the two main heat fluxes transferred to the workpiece are conduction heat flux and electric heat flux. These two heat fluxes are related to the distribution of temperature and current density. The current density distributions at the x = 0 section are shown in Figure 6. When the time is 0.0252 s, the welding arc is at the transient state, as stated in Figure 5. At this time, the welding arc profile and the current flow path are close to the normal welding arc. In Figure 6b, most of the current flows through the bottom of the welding groove, and there is almost no current flow through the sidewall. When the MOA deflects to the sidewall, part of the electrons flow through the groove sidewall to the tungsten electrode tip, and the remaining electrons still flow through the bottom. As such, the electric heat transferred to the sidewall will be increased, and that transferred to the bottom will be decreased. Although the axis of the welding arc temperature has already deflected to the sidewall in the simulation case, most of the welding current still flows through the bottom. The change of the electric heat is asynchronous with the change of conduction heat.

The total heat flux distribution on the welding groove faces is shown in Figure 7. At this time, the heat flux is mainly distributed at the bottom and the left sidewall, and the heat flux at the right sidewall is so small that it can be ignored. The maximum heat flux is still at the bottom, even though the welding arc axis is already deflected to the sidewall. The reason is the current density at the sidewall is still small and so is the electric heat flux. The distribution shape of the heat flux is close to an oval at the bottom, and it is triangular at the left side. The heat flux near the corner of the groove is small, which could lead to a lack of fusion here.

The heat fluxes along the centerlines of the bottom and left sidewall are also shown in Figure 7b,c, and the heat flux with no LMF is provided as a reference. At the bottom centerline, compared with the heat flux under the normal welding arc, the maximum value of heat flux decreases by almost half but decays slowly along the deflecting direction of the welding arc. At the sidewall, the increment of the maximum heat flux is more than 10 times. The maximum value appears at the position about 1 mm away from the corner, and there is also a slowdown area. It can be concluded that, compared with the normal welding arc, the heat flux is no longer concentrated to one point but is scattered along the bottom and sidewall, which is helpful to forming a uniform penetration welding pool at the bottom and increasing penetration at the sidewall.

### 3.2. Fluid Flow of Molten Pool

The fluid flow of the molten pool directly influences the final welding bead. The flow pattern is mainly dependent on the shear stress, arc pressure, surface tension, Marangoni force, gravity and buoyancy. The first two forces come from the welding arc and are influenced greatly by the MOA. The velocity of the arc plasma, pressure and the shear stress on the weld pool surface at two times is shown in Figure 8. The arc plasma flows to the groove sidewall rapidly and then changes direction to flow out. Under the influence of the arc plasma, the arc pressure and shear stress are mainly distributed on the sidewall, and the maximum value also appears at the sidewall. The values of arc pressure and shear stress along the bottom are relatively small. At the groove bottom, the location with high arc pressure is depressed, and the other position is put forward correspondingly. The high arc pressure appearing at the sidewall could lead to undercut defects, so from this point, the deflecting amplitude should be controlled. The shear stress forces the molten metal to flow along the tangential direction of the surface. Therefore, under the effect of the arc shear stress, the molten metal at the bottom will flow to the corner and then flow upward at the sidewall.

The fluid flow in the molten pool is shown in Figure 9. In Figure 9a the MOA deflects to the left side, and in Figure 9b the MOA deflects to the right side. The flow pattern can be divided into two parts; one part is contacting with the deflected arc, and the other part is far away from the deflected arc. In the former part, the molten metal at the bottom flows down and then flows up to the sidewall under the effect of the arc pressure and shear stress. The liquid metal at the sidewall flows up along with the arc plasma and then changes its direction to flow down after contacting with the solid workpiece. In the latter part, the liquid metal is interconnected from the bottom and the sidewall, and they form a circular movement; the liquid metal at the sidewall flows down to the bottom and then changes its direction to flow back.

According to the heat flux distribution shown in Figure 7, it can be concluded that the temperature of the bottom of the workpiece is the highest, the temperature at the sidewall is lower and the temperature at the corner is the lowest. Under the above flow pattern, in the part contacting the deflected arc, the molten metal at the bottom can transfer heat to the sidewall; in the other part, the liquid metal at the sidewall transfers heat to the corner. This flow pattern can compensate for the distribution difference of heat flux and is helpful for obtaining uniform welding beads.

### 3.3. Discussion

This model was developed based on some assumptions; for example, the distribution of LMF was uniform, and the metal vapor was not considered. These assumptions had an influence on the precision of the predicted results. For example, the magnetic density will decrease along the arc column in practice, which can influence the profile of the deflecting arc and then influence the heat transfer. If the metallic vapor diffuses to the welding arc zone, the properties of the arc plasma will be changed, such as the thermal conductivity, magnetic conductivity, radiation coefficient and so on. These variables have effects on the temperature, current density and velocity of the welding arc. Thus, if the magnetic excitation equipment and the metal vapor arc are included, the accuracy of the model will be improved. In addition, the surface tension and its gradient of temperature of the base metal have a large influence on the flow pattern of the liquid metal. In this model, the surface tension gradient of temperature is negative; the liquid metal flows to the zone with lower temperature under the effect of surface tension. If the active element content or the kind of metal is changed, the predicted flow pattern in the welding pool will also be changed.

## 4. Validation of the Model

Wang et al. [16] used the same welding system employed in this work to measure the welding arc pressure by piezoresistive pressure sensor, but applied water-cooled copper as the anode and drilled four holes at the bottom and sidewall of the narrow groove. This experimental case was similar to our simulation model when the welding pool was not formed. The dimension of the workpiece was the same as our model. Their magnetic field generator had two magnetic poles, so the magnetic field density was nearly uniform in the arc column; this distribution of LMF was also similar to our assumption. Therefore, the experimental pressure could be used to verify our model. The positions of the four points are shown in Figure 10. Point 1 is located in the bottom center and is 4 mm away from point 2; the distances from the corner to point 3 and 4 are 1 mm and 2 mm, respectively. The welding parameters were: the flow rate of shielding gas was 25 L/min, the diameter of the electrode was 3.2 mm; the current was 240 A; the magnetic flux density was 6 mT. Because the copper was not melted in the experiment, the simulation values were selected at a time when the workpiece was not melted. Both the experimental and simulation results obtained the biggest value at point 2, followed by point 3, and the pressure was low at points 1 and 4. The calculated arc pressures matched well with the measured values, especially at points 1 and 2, and the trend of arc pressure was the same.

Sun et al. [14] welded stainless steel 304 based on the same experimental system. The detailed parameters were: a magnetic field intensity of 9 mT; frequency of 10 Hz; welding current of 200 A; welding speed of 60 mm/min. The experimental results were compared with the calculated weld bead dimensions. The results are shown in Figure 11 and Table 3. The calculated weld width is 14.2 mm, and the weld depth is 1.8 mm. Compared with the experimental results, the simulation error is smaller than 9%.

## 5. Conclusions

(1)A unified mathematical model was developed for MOA-NG-GTA welding, including the magnetic oscillation arc, weld pool, the workpiece and the heat and mechanical coupling between them. The model was validated by comparison with the experimental results.(2)The influence of the external alternating longitudinal magnetic field on the formation of welds was complicated. Although it was only added to the arc column, the current density of the welding zone, the heat flux and the forces along the pool surface were all changed, which affected the formation of the weld bead.(3)The prediction accuracy of the heat and mechanical properties of the deflecting arc had a key effect on the simulation of the weld formation. Thus, the factors that affected the welding arc will be considered in future work, such as the distribution of the external magnetic field and the metal vapor.

## Figures and Tables

**Figure 1 materials-13-05799-f001:**
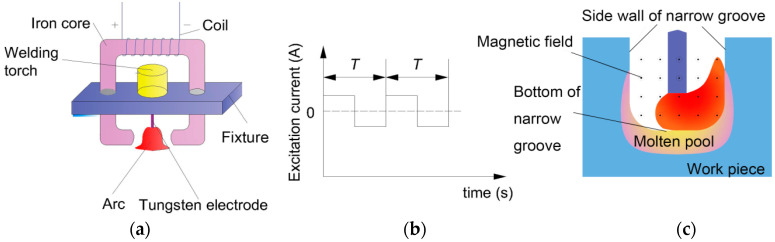
Schematic of magnetic oscillation arc narrow gap gas tungsten arc welding (MOA-NG-GTAW): (**a**) Magnetic field generator; (**b**) Waveform of magnetic excitation current; (**c**) Deflecting welding arc.

**Figure 2 materials-13-05799-f002:**
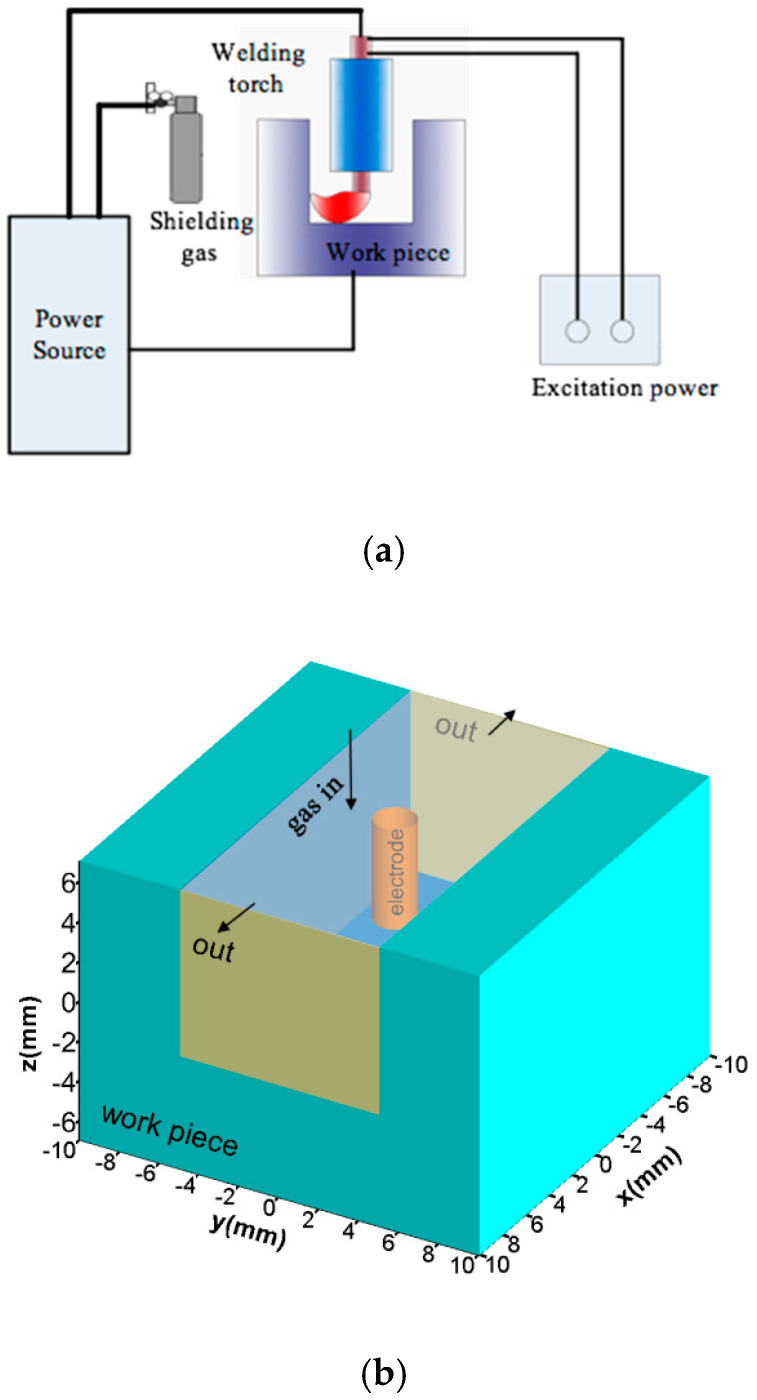
Experimental system and computational domain of MOA-NG-GTAW: (**a**) Experimental system; (**b**) Simulation domain.

**Figure 3 materials-13-05799-f003:**
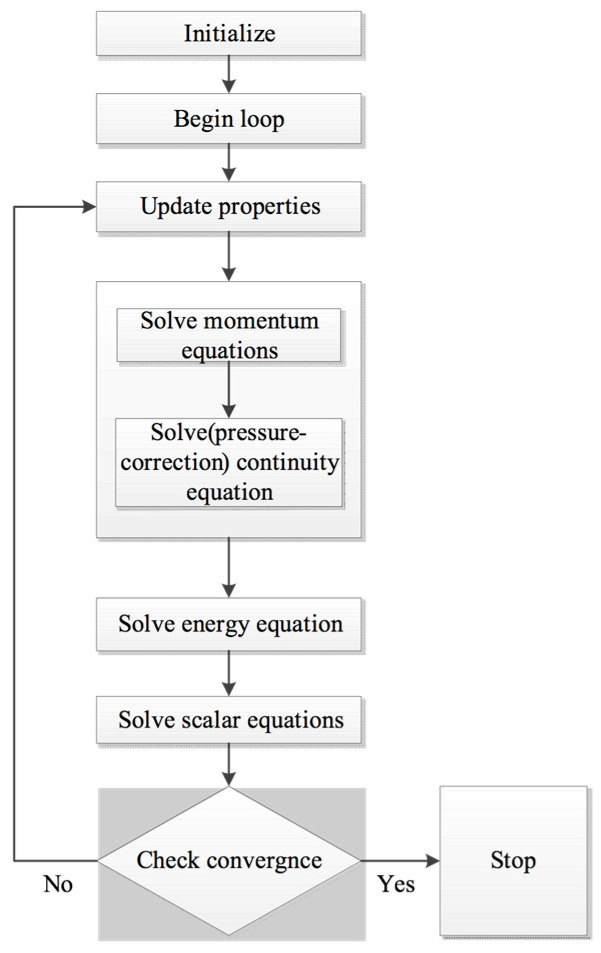
Flow chart of the calculation.

**Figure 4 materials-13-05799-f004:**
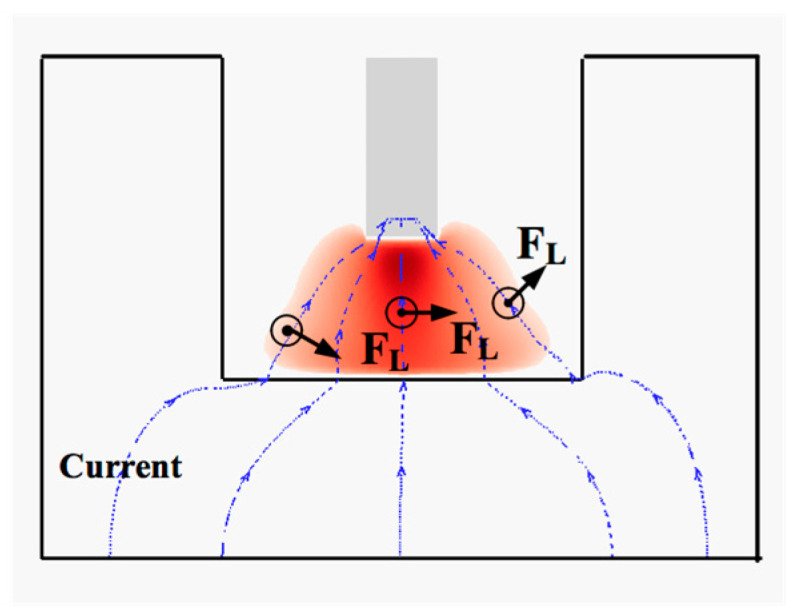
Distribution of Lorentz force from the extra longitudinal magnetic field (LMF).

**Figure 5 materials-13-05799-f005:**
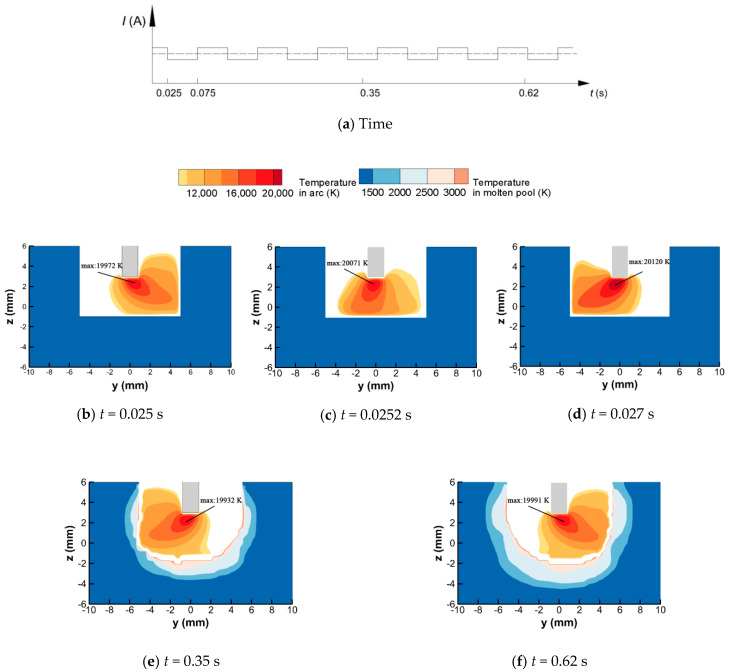
Distribution of MOA temperature at section x = 0 at different times: (**a**) time; (**b**) *t* = 0.025 s; (**c**) *t* = 0.0252 s; (**d**) *t* = 0.027 s; (**e**) *t* = 0.35 s; (**f**) *t* = 0.62 s.

**Figure 6 materials-13-05799-f006:**
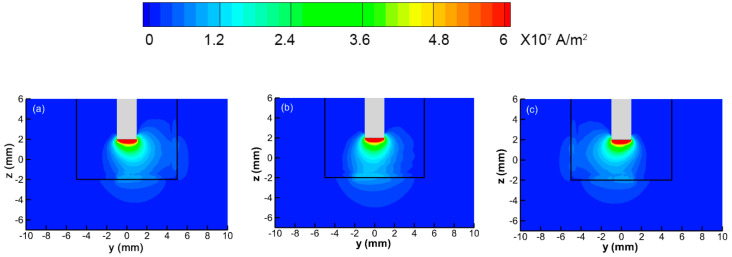
Current density at cross section (x = 0): (**a**) t = 0.0250 s; (**b**) t = 0.0252 s; (**c**) t = 0.027 s.

**Figure 7 materials-13-05799-f007:**
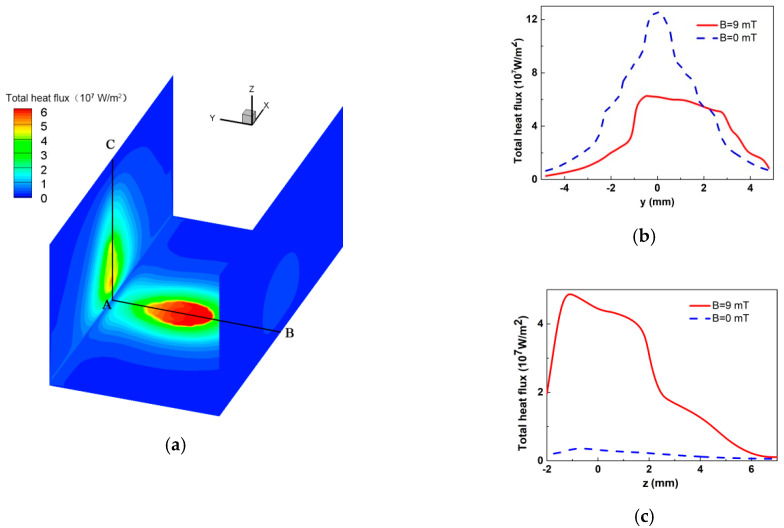
Total heat flux distribution: (**a**) Distribution of heat flux at groove faces; (**b**) Heat flux distribution along centerline AB; (**c**) Heat flux distribution along centerline AC (t = 0.027 s).

**Figure 8 materials-13-05799-f008:**
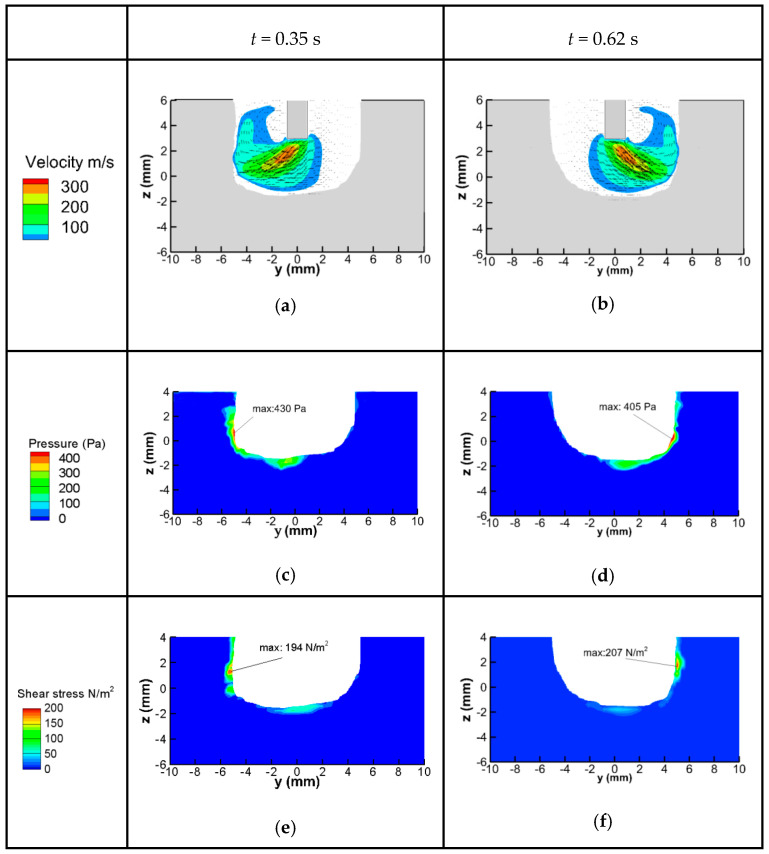
Distribution of welding arc velocity, the arc pressure and shear stress at x = 0 section: (**a**,**b**) Welding arc velocity; (**c**,**d**) Arc pressure; (**e**,**f**) Shear stress.

**Figure 9 materials-13-05799-f009:**
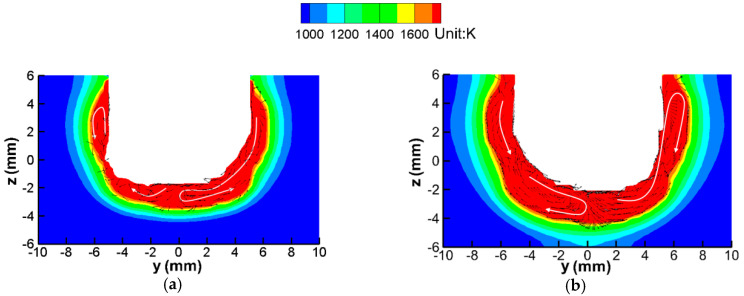
Fluid flow of the molten pool on cross section: (**a**) t = 0.35 s; (**b**) t = 0.62 s.

**Figure 10 materials-13-05799-f010:**
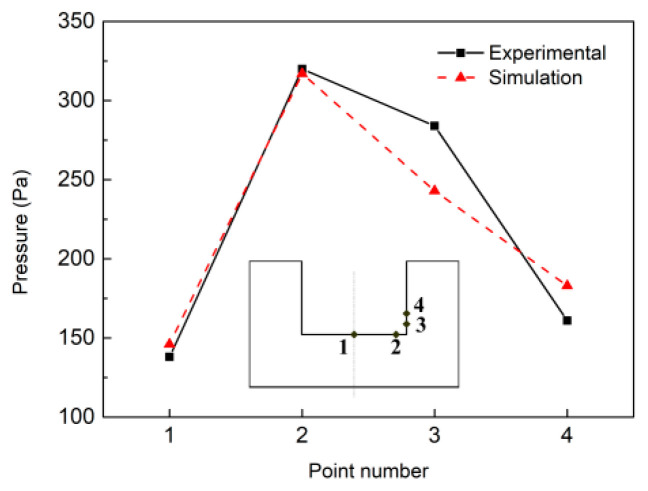
Comparison between simulated and experimental arc pressure of four points, experimental data from [7].

**Figure 11 materials-13-05799-f011:**
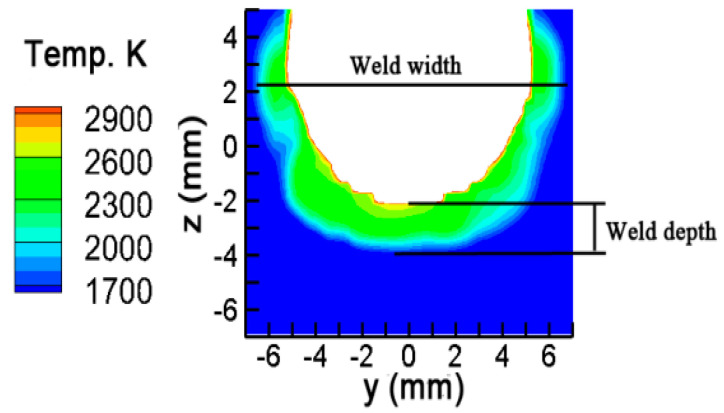
Comparison between simulated and experimental weld bead, experimental data from [6].

**Table 1 materials-13-05799-t001:** The external boundary conditions.

Boundary	*T* (K)	*V* (m/s)	*Φ* (V)	*A*
Gas Inlet	500	*v_x_* = *v_y_* = 0, *v*_z_ = *v*_g_	$\frac{\partial \phi }{\partial \vec{n}}=0$	$\frac{\partial \vec{A}}{\partial \vec{n}}=0$
Tungsten Electrode Tip	3000	-	$-\sigma \frac{\partial \phi }{\partial \vec{n}}=j_{z}$	$\frac{\partial \vec{A}}{\partial \vec{n}}=0$
Tungsten Surface	1000	-	$\frac{\partial \phi }{\partial \vec{n}}=0$	$\frac{\partial \vec{A}}{\partial \vec{n}}=0$
Outlet	1000	-	$\frac{\partial \phi }{\partial \vec{n}}=0$	0
External Faces of the Workpiece	$k\nabla T-\varepsilon \alpha T^{4}$	-	*ϕ* = 0	$\frac{\partial \vec{A}}{\partial \vec{n}}=0$

**Table 2 materials-13-05799-t002:** Major physical properties of SUS304 stainless steel used in this model.

Nomenclature (Symbol)	Value
Freezing Point (*T*_liquidus_)	1670 K
Melting Point (*T*_solidus_)	1727 K
Density (*ρ*)	7200 kg m^−3^
Electric Wonductivity (σ)	7.7 × 10^5^ S/m
Surface Tension Coefficient	1.2 N m^−1^
Surface Tension Temperature Gradient ∂γ/∂T	1 × 10^−4^ N m^−1^ K^−1^
Work Function(*φ_w_*)	4.65 eV

**Table 3 materials-13-05799-t003:** Comparison between measured and predicted welds.

	Measure [6]	Prediction	Deviation
Weld Width (mm)	14.6	14.2	2.8%
Weld Depth (mm)	2	1.8	8.7%

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
