# Peer review of "Numerical Analysis of the Heating Characteristics of Magnetic Oscillation Arc and the Fluid Flow in Molten Pool in Narrow Gap Gas Tungsten Arc Welding"

_materials, 2020, doi:10.3390/ma13245799_

Round 1

Reviewer 1 Report

The paper presents the study regarding simulation of processes that took place in molten welding pool. The paper presents the results which focus on the magnetic oscillation arc in NG-GTAW welding process.

The paper is well written. The introduction and research (simulation) part are presented clearly and describes sufficiently the object of the work. Conclusions are clear and derives from the research (simulation) material. 

Congratualltions for a good work.

Author Response

The authors are very grateful to the reviewer for the positive affirmation of this work. We will continue efforts to do better

Reviewer 2 Report

1) Rewrite the motivation for this study.

2) Explain more about the methods used in this study.

3) Rewrite conclusion as per guidelines.

Reviewer 3 Report

The paper deals with Numerical analysis of the heating characteristics of magnetic oscillation arc and the fluid flow in molten pool in narrow gap gas tungsten arc welding .

The structure of the scientific report is good and well-understood. The aim is clarified. The introduction summarizes well.

Bibliographic references are appropriate. The author mentioned 28 literatures according to the reference list. These references are from the last 10 years, from prestigious journals.

The reviewer suggests to accept in the present form for publication.

Author Response

The authors are very grateful to the reviewer for the positive comments. We will continue efforts to do better. 

Reviewer 4 Report

General Comments. The manuscript needs to be carefully checked by an expert in scientific English for engineering materials. While the manuscript is clearly written, there are places where editorial improvements in the English are needed.

In Section 2, be careful that all of the necessary references are included. In addition, while the manuscript is well-referenced, many readers of the journal Materials will not have a strong background in the detailed theoretical concepts and mathematical equations that are being presented, and some further references of a general background nature would be worthwhile.

The Results and Discussion, Section 3, needs an additional paragraph that provides comments about the limitations of the study and contains suggestions for future research. For example, the experimental verification of the mathematical modeling was validated with a stainless steel as the workpiece material. For what other materials does the MOA-assisted NG-GTAW technique have engineering applicability, and would any difficulties be expected with modeling this welding technique for these other materials?

Reviewer 5 Report

The research ”Numerical analysis of the heating characteristics of magnetic oscillation arc and the fluid flow in molten pool in narrow gap gas tungsten arc welding ” authored by Xiaoxia Jian, Xing Yang, Jingqian Li and Hebao Wu is interesting and it is in the area of research of the journal. The objective of this study is to investigate the heating characteristics of welding arc and fluid flow in the molten pool in magnetic oscillation arc narrow gap gas tungsten arc welding. Because both the magnetic oscillation arc and the fluid flow in molten pool are relating to multi-physics coupling, the mathematical simulation method was selected. However, the paper is not presenting an experimental part – only the validation of the mathematical model with the literature research studies – e.g. J. Wang, Q. Sun, T. Zhang, S. Zhang, Y. Liu, J. Feng. Arc characteristics in alternating magnetic field assisted narrow gap pulsed GTAW. J. Mater. Process. Technol. 2018, 254, 254-264 or V. Y. Belous. Conditions for formation of defect-free welds in narrow-gap magnetically controlled arc welding of low titanium alloys. The Paton Weld. J. 2011, 3,16 -18. This link is not clear and does not have description (studied model vs experimental results validity). Also the mathematical approach is poor explained and the convergence of the solution is not revealed – only assumption of the theoretical model are presented. I do recommend authors to revise the paper and clear describe the validation for the theoretical and numerical model (even use of the existing research results).

Round 2

Reviewer 5 Report

The authors have done a major revision of the paper. All the required elements were clear explained and completed. The paper can be accepted in the present form.